

# Coexistence of valence-bond formation and topological order in the frustrated ferromagnetic $J_1$-$J_2$ Chain

**Cliò Efthimia Agrapidis**[1]⋆, **Stefan-Ludwig Drechsler**[1],
**Jeroen van den Brink**[1,2] **and Satoshi Nishimoto**[1,2]

**1** Institute for Theoretical Solid State Physics, IFW Dresden, Dresden, Germany
**2** Department of Physics, Technical University Dresden, Dresden, Germany

⋆ c.agrapidis@ifw-dresden.de

## Abstract

Frustrated one-dimensional (1D) magnets are known as ideal playgrounds for new exotic quantum phenomena to emerge. We consider an elementary frustrated 1D system: the spin-$\frac{1}{2}$ ferromagnetic ($J_1$) Heisenberg chain with next-nearest-neighbor antiferromagnetic ($J_2$) interactions. On the basis of density-matrix renormalization group calculations we show the existence of a finite spin gap at $J_2/|J_1| > 1/4$ and we find the ground state in this region to be a valence bond solid (VBS) with spin-singlet dimerization between third-neighbor sites. The VBS is the consequence of spontaneous symmetry breaking through order by disorder. Quite interestingly, this VBS state has a Affleck-Kennedy-Lieb-Tasaki-type topological order. This is the first example of a frustrated spin chain in which quantum fluctuations induce gapped topological order.

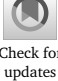
# 1 Introduction

The one-dimensional quantum world of spin-chain systems connects some of the most advanced concepts from many-body physics, such as integrability and symmetry-protected topological order [1], with the measurable physical properties of real materials. An example is the presence of the Haldane phase [2] in spin-1 chains, which is a topological ground state protected by global $Z_2 \times Z_2$ symmetry [3,4]. On the other hand frustrated magnets, in which a macroscopic number of quasi-degenerate states compete with each other, are an ideal playground for the emergence of exotic phenomena [5]. For instance, the interplay of frustration and fluctuations leads to unexpected condensed matter orders at low temperatures by spontaneously breaking of either a continuous or discrete symmetry, i.e., order by disorder [6]. One of the simplest systems that shares both these features – geometric frustration and one-dimensionality – is the so-called $J_1$-$J_2$ chain, the Hamiltonian of which is given by

$$H = J_1 \sum_i \mathbf{S}_i \cdot \mathbf{S}_{i+1} + J_2 \sum_i \mathbf{S}_i \cdot \mathbf{S}_{i+2}, \tag{1}$$

where $\mathbf{S}_i$ is spin-$\frac{1}{2}$ operator at sites $i$, $J_1$ is nearest-neighbor (NN) and $J_2$ is next-nearest-neighbor (NNN) interactions. This chain system can be also represented as a zigzag ladder [Fig. 1(a)] or a diagonal ladder [Fig. 1(b)-(d)]. The NNN interaction is assumed to be antiferromagnetic (AFM), i.e. $J_2 > 0$, inducing geometrical frustration. The frustration is parametrised as $\alpha = J_2/|J_1|$. The magnetic properties are quite different between the cases of ferromagnetic (FM) $J_1 < 0$ and AFM $J_1 > 0$, where we denote the systems as "FM $J_1$-$J_2$ chain" and "AFM $J_1$-$J_2$ chain", respectively. In this paper, we restrict ourselves to the FM $J_1$-$J_2$ chain, which is used as a standard magnetic model for quasi-one-dimensional edge-shared cuprates such as $Li_2CuO_2$ [7], $LiCuSbO_4$ [8], $LiCuVO_4$ [9], $Li_2ZrCuO_2$ [10], $Rb_2Cu_2Mo_3O_{12}$ [11] and $PbCuSO_4(OH)_2$ [12]. Especially, multi-magnons bound state [13] and multipolar ordering [14] under magnetic field have been established both theoretically and experimentally in this context.

The ground state of the AFM $J_1$-$J_2$ chain is well understood [15–17], assisted by the exact solution of the Majumdar-Ghosh model for $\alpha = 0.5$ [18]; but surprisingly the ground and excited state properties of the FM $J_1$-$J_2$ chain are still not completely identified. It is known that a phase transition occurs at $\alpha = \frac{1}{4}$ [19,20] from a FM to an incommensurate spiral state [21,22] with dimerization order [23], but the quantitative estimation of spin gap (if it exists) and its numerical confirmation have been a long standing challenge - so far there is only a field-theoretical predictions of an exponentially small spin gap for $\alpha \gtrsim 3.3$ [24,25].

Our aim is to determine the ground state and spin gap of the FM $J_1$-$J_2$ chain. To this end, we calculated various quantities including spin gap, string order parameter, several dimerization order parameters, dimer-dimer correlation function, spin-spin correlation function, and entanglement entropy using the density-matrix renormalization group (DMRG) technique [26]. First, we verify the existence of a finite spin gap at $\alpha > \frac{1}{4}$ and find its maximum around $\alpha \simeq 0.6$. Next, we show that the ground state is a valence bond solid (VBS) state with spin-singlet formations between third-neighbor sites (which we refer to as the "$\mathcal{D}_3$-VBS state"), which leads to the finite spin gap. The leading mechanism for the emergence of this ordered state is magnetic frustration, which is characterized by the presence of strong quantum fluctuations: while the classical ground state is highly degenerate, quantum fluctuations in the system lift this degeneracy with formation of FM dimers and valence bonds, thus we are observing the formation of *order by disorder*. Remarkably, this VBS state is associated with an Affleck-Kennedy-Lieb-Tasaki (AKLT) [27]-like topological hidden order. While there exist examples of order by disorder in quantum chains (e.g. Majumdar-Ghosh model [18]), we are not aware of previous example of *topological order by disorder*. We support the topological nature of the $\mathcal{D}_3$-VBS state by com-

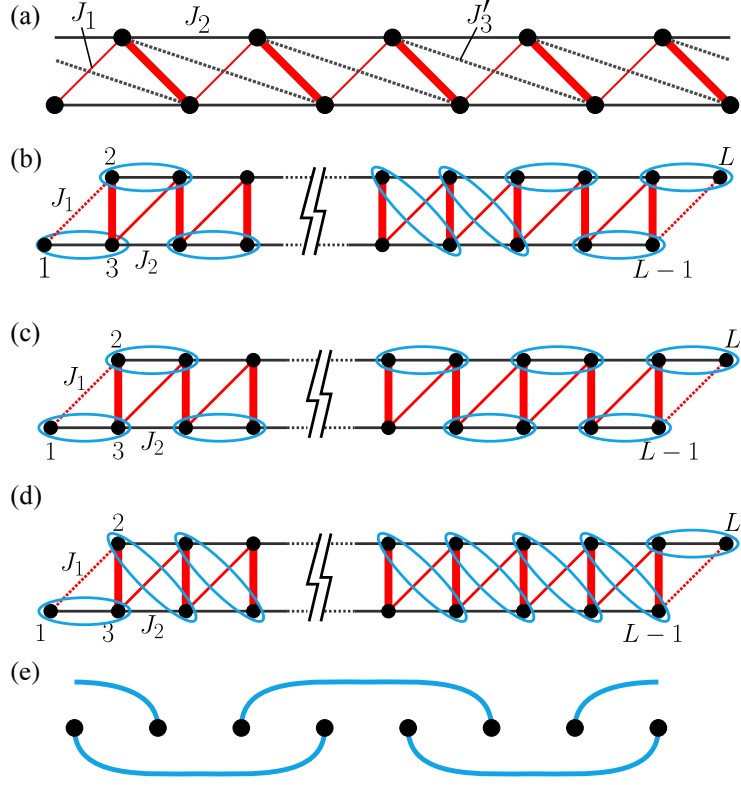

Figure 1: (a) Lattice structure of the $J_1$-$J_2$ chain (at $J_3' = 0$) as a zigzag ladder. The $J_1$ chain is shown in red. Thick lines represent spin-triplet dimers, which are spontaneously formed in the VBS state. Dotted lines show the third-neighbor $J_3'$ bonds (see text). (b)(c)(d) Three candidates for the VBS ground state of the FM $J_1$-$J_2$ chain. A red thick line represents an effective $S = 1$ site as a spin-triplet pair of two spin-$\frac{1}{2}$ sites, a blue ellipse represents a spin-singlet pair, i.e., valence bond. The dashed $J_1$ bonds at the chain edges are set to be zero in most of our calculations. (e) Schematic picture of the third-neighbor VBS ground state ("$\mathcal{D}_3$-VBS state") of the FM $J_1$-$J_2$ chain.

puting the entanglement spectra (ES) of the system. We confirm the robustness of the $\mathcal{D}_3$-VBS state by considering an adiabatic connection of the ground state to the enforced third-neighbor dimerized state.

## 2 Methods

We employ the DMRG method, which is one of the most powerful numerical techniques for studying 1D quantum systems. Open boundary conditions (OBC) are applied unless stated otherwise. Besides, both edged $J_1$'s (denoted as $J_1^{\text{edge}}$) are taken to be zero in the open chain. This has an important physical implication which will be clarified in the following. This enables us to calculate ground-state and low-lying excited-state energies, as well as static quantities, quite accurately for very large systems. This puts us in the position to carry out an accurate finite-size-scaling analysis to obtain energies and quantities in the thermodynamic limit. We keep up to $m = 6000$ density-matrix eigenvalues in the renormalization procedure. Moreover, several chains with length up to $L = 800$ are studied to perform finite size scaling. This way, we are able to obtain accurate results with error in the energy $\Delta E/L < 10^{-11}$. In some cases we

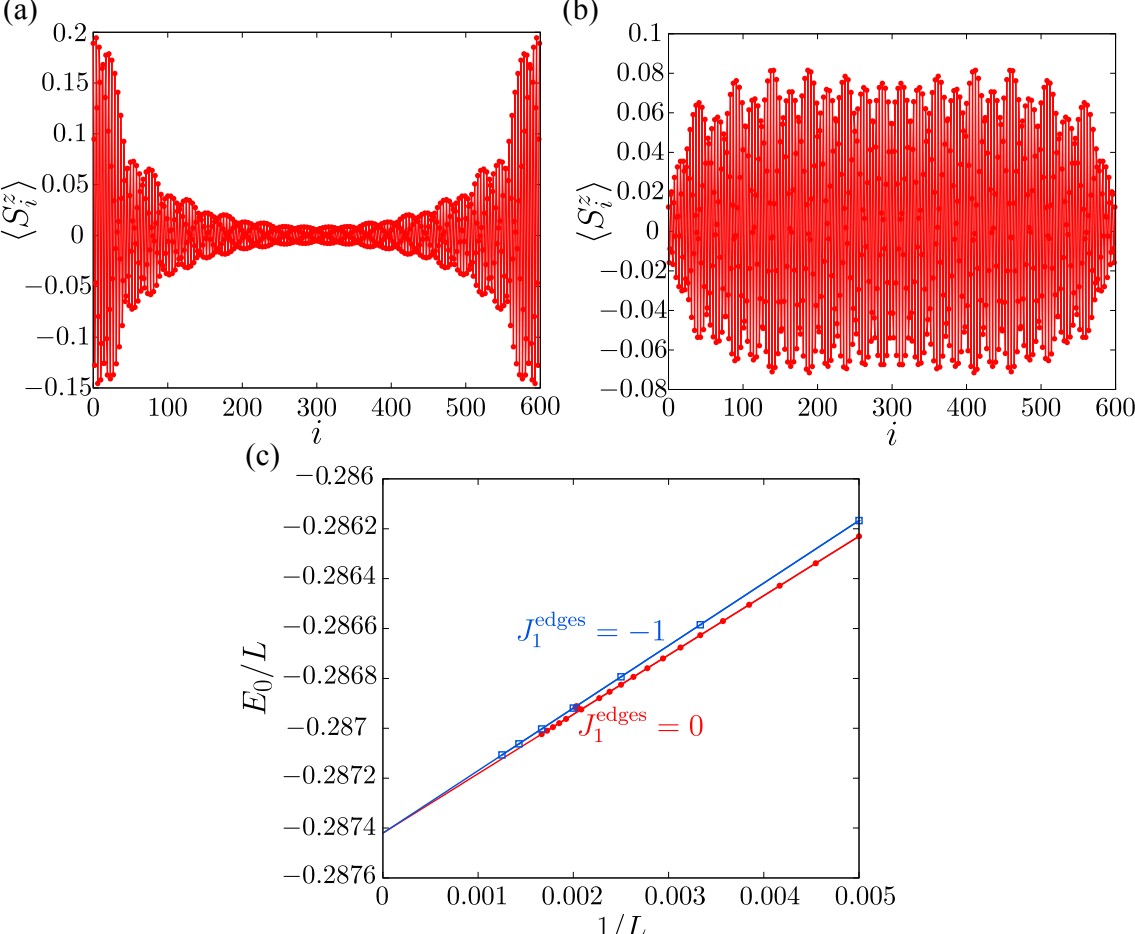

Figure 2: Expectation value of the $z$-component of local spin $\langle S_i^z \rangle$ in the first-excited triplet state (total $S^z = 1$) as a function of site position $i$ at $\alpha = 0.6$ with $L = 600$ for (a) $J_1^{\text{edge}} = -1$ and (b) $J_1^{\text{edge}} = 0$. (c) Finite-size scaling of the lowest-state energy with total $S^z = 0$ for $J_1^{\text{edge}} = -1$ and $J_1^{\text{edge}} = 0$ at $\alpha = 0.6$. A linear fitting is performed in both cases.

study larger systems up to $L = 3000$ to estimate the decay length of the spin-spin correlation function and entanglement entropy.

## 3 Spin gap

Although the existence of a tiny spin gap was predicted by the field-theoretical analyses [24, 25], it has not been numerically detected so far. In our DMRG calculations, the spin gap $\Delta$ is defined as the energy difference between the singlet ground state and the triplet first excited state:

$$\Delta(L) = E_0(L, S^z = 1) - E_0(L, S^z = 0); \quad \Delta = \lim_{L \to \infty} \Delta(L), \tag{2}$$

where $E_0(L, S^z)$ is the ground state energy of a system of size $L$ and total spin $z$-component $S^z$. As mentioned above, we set $J_1^{\text{edge}} = 0$; otherwise, one cannot measure correctly the excitation energy for the bulk system. As shown below, our system is spontaneously dimerized along the FM $J_1$ chain. By regarding the ferromagnetically dimerized NN bond as a $S = 1$ site, the system can be considered as a $S = 1$ Heisenberg chain. In fact, this setting $J_1^{\text{edge}} = 0$ corresponds to

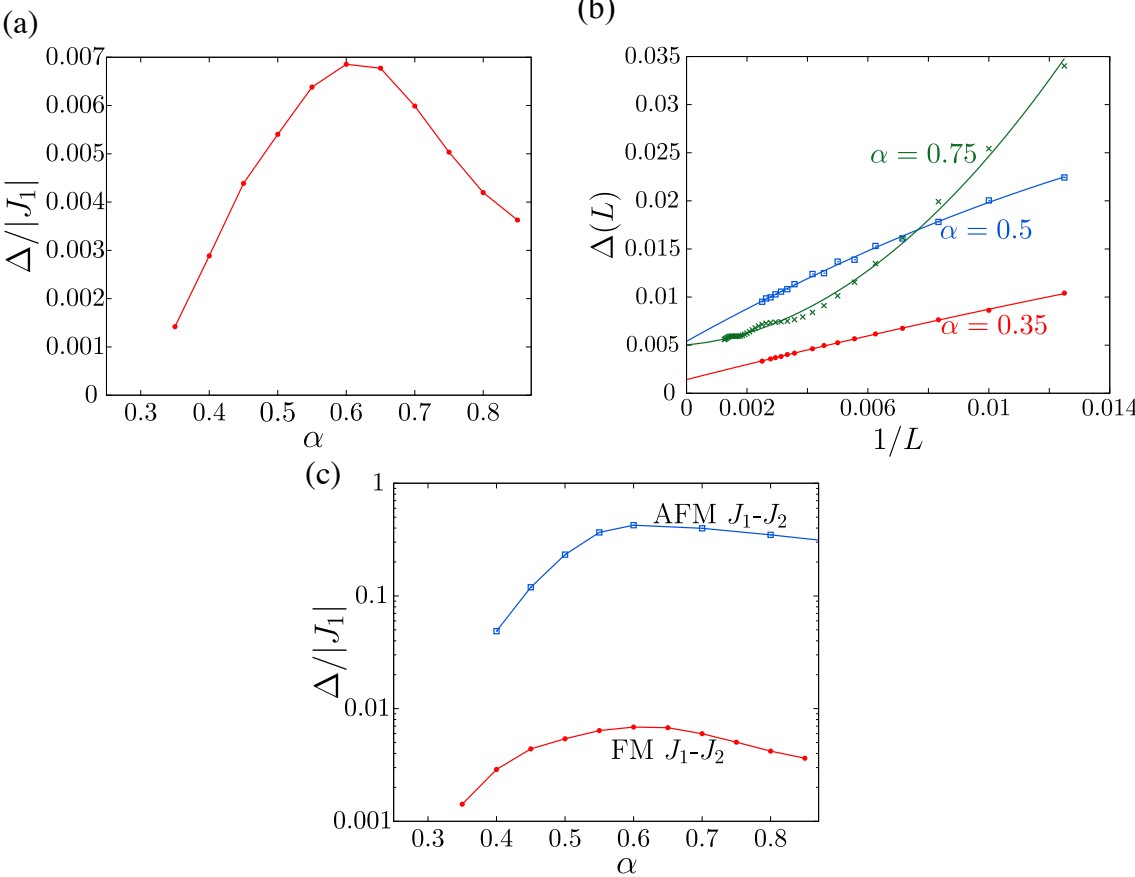

Figure 3: (a)Spin gap $\Delta/|J_1|$ of the $J_1$-$J_2$ chain as a function of the degree of frustration $\alpha$. (b) Examples of finite size scaling of the spin gap for $\alpha = 0.35$ (red line), $\alpha = 0.5$ (blue line) and $\alpha = 0.75$ (green line). (c) Comparison between the gaps of the FM $J_1 - J_2$ and the AFM $J_1$-$J_2$ chain on a semilog scale.

an explicit replacement of $S = 1$ spin at each end by $S = \frac{1}{2}$ spin in the $S = 1$ Heisenberg open chain. It is known that this procedure is necessary to numerically calculate the Haldane gap as a singlet-triplet excitation defined by Eq.(2) because a $S = \frac{1}{2}$ degree of freedom appears as an unpaired (nearly) free spin at both edges, i.e., so-called edge spin state, in the $S = 1$ Heisenberg open chain. The appearance of edge spin states is a definite signature of the Haldane state. To illustrate the presence of edge spin states in our model, we plot the expectation value of the local spin $z$-component, i.e. $\langle S_i^z \rangle$, in the $S^z = 1$ first-excited triplet state as a function of site position $i$ at $\alpha = 0.6$ for $L = 600$. As shown in Fig. 2(a), when we naively keep $J_1^{\text{edge}} = -1$, the spin flipped from the singlet ground state (spinon) is mostly localized around the chain edges. It resembles the fact that a residual $S = 1/2$ edge spin (out of a valence bond) in the 1D $S = 1$ Heisenberg model can be flipped without energy cost. In this case, the excitation energy, i.e. the spin gap, is zero or significantly underestimated. It thus prevents us from estimating the bulk spin gap correctly. Whereas in the case of $J_1^{\text{edge}} = 0$, the flipped spin is distributed inside the system as seen in Fig. 2(b). Therefore, this setting of $J_1^{\text{edge}} = 0$ enables us to obtain the spin gap after an extrapolation of the singlet-triplet excitation energy to the thermodynamic limit.

Fig. 3(a) shows the spin gap in the thermodynamic limit as a function of $\alpha$. For information, we present three examples of finite-size scaling analysis for the spin gap in Fig. 3(b). We performed second-order polynomial fitting for all values of $\alpha$. For $\alpha \geq 0.6$, larger system sizes

up to $L = 800$ were taken into account due to the oscillations of the data point reflecting the incommensurate structure. For $\alpha > 0.85$ the oscillations become a crucial problem and we could not perform a reasonable fitting. The spin gap of the FM $J_1$-$J_2$ chain is compared to that for the AFM $J_1$-$J_2$ chain in Fig. 3(c). For the FM $J_1$-$J_2$ chain a finite spin gap is clearly observed in a certain $\alpha$ region, although it is about two orders of magnitude smaller than that for the AFM $J_1$-$J_2$ chain. The spin gap seems to grow continuously from $\alpha = \frac{1}{4}$ reaching its maximum $\Delta \simeq 0.007|J_1|$ around $\alpha \simeq 0.6$, which is within *the most highly-frustrated region*. This already suggests that the origin of the spin gap is a frustration-induced long-range order, and the result of *order by disorder*.

We here check to be sure that the artificial setting $J_1^{\text{edge}} = 0$ does not change the ground state. To study it, we compare the lowest energies at $\alpha = 0.6$ for the two different values of $J_1^{\text{edge}}$ in Fig. 2(c) as a function of $1/L$. We see that at finite $L$ the energy for $J_1^{\text{edge}} = 0$ is rather lower than that for $J_1^{\text{edge}} = -1$. Nevertheless, they coincide perfectly in the thermodynamic limit ($1/L = 0$); a linear fitting yields $E_0/L = -0.2874202246$ for $J_1^{\text{edge}} = -1$ and $E_0/L = -0.2874200731$ for $J_1^{\text{edges}} = 0$. This means that the bulk ground state does not depend on the choice of $J_1^{\text{edge}}$.

Additionally, it would be interesting to mention the relation between edge spin states and spinon excitations. Since the spin gap is very small in our system, the spinons are expected to be nearly deconfined. With setting $J_1^{\text{edge}} = -1$, the system exhibits spin edge states; thus, a spinon is created at the system edges as an edge spin-$\frac{1}{2}$ excitation in the total $S^z = 1$ state [see Fig. 2(a)]. Typically, the Friedel oscillation decays quickly (with decay length of the order of 1) from the edges in a Haldane gapped system. If the edge spin-$\frac{1}{2}$ is completely free like in the AKLT state, the decay length is 0. However, in our system, it decays very slowly and the amplitude seems to be still sizable even around the system center for $L = 600$. The slow decay of the Friedel oscillation clearly indicates nearly complete deconfinement of spinons. This is also consistent with an exponential decay of the spin-spin correlation with very large decay length, $\xi \sim 50$ ($\alpha \sim 0.6$) at the minimum.

## 4  Valence Bond Solid

Having established the existence of a finite spin gap for $\alpha > \frac{1}{4}$, we investigate a possible mechanism leading to it. It is known that a spontaneous FM dimerization is driven along $J_1$ bonds [24, 25] and an emergent effective spin-1 degrees of freedom is created with the dimerized two spin-$\frac{1}{2}$'s [23]. If the system (1) can be mapped onto a $S = 1$ Heisenberg chain, the finite spin gap might be interpreted as a Haldane gap with a VBS state [27]. However, it is nontrivial whether an arbitrary set of valence bonds, i.e., resonating valence bonds forming in different directions, between the neighboring effective $S = 1$ sites leads to a finite spin gap [see Fig. 1(b)]. To investigate the stability of VBS state, we examine the string order parameter [28]:

$$\mathcal{O}^z_{\text{string}} = -\lim_{|k-j|\to\infty} \langle (S^z_k + S^z_{k+1}) \exp(i\pi \sum_{l=k+2}^{j-1} S^z_l)(S^z_j + S^z_{j+1})\rangle. \tag{3}$$

For our system (1), Eq.(3) can be simplified as

$$\mathcal{O}^z_{\text{string}} = -\lim_{|k-j|\to\infty} (-4)^{\frac{j-k-2}{2}} \langle (S^z_k + S^z_{k+1}) \prod_{l=k+2}^{j-1} S^z_l(S^z_j + S^z_{j+1})\rangle \tag{4}$$

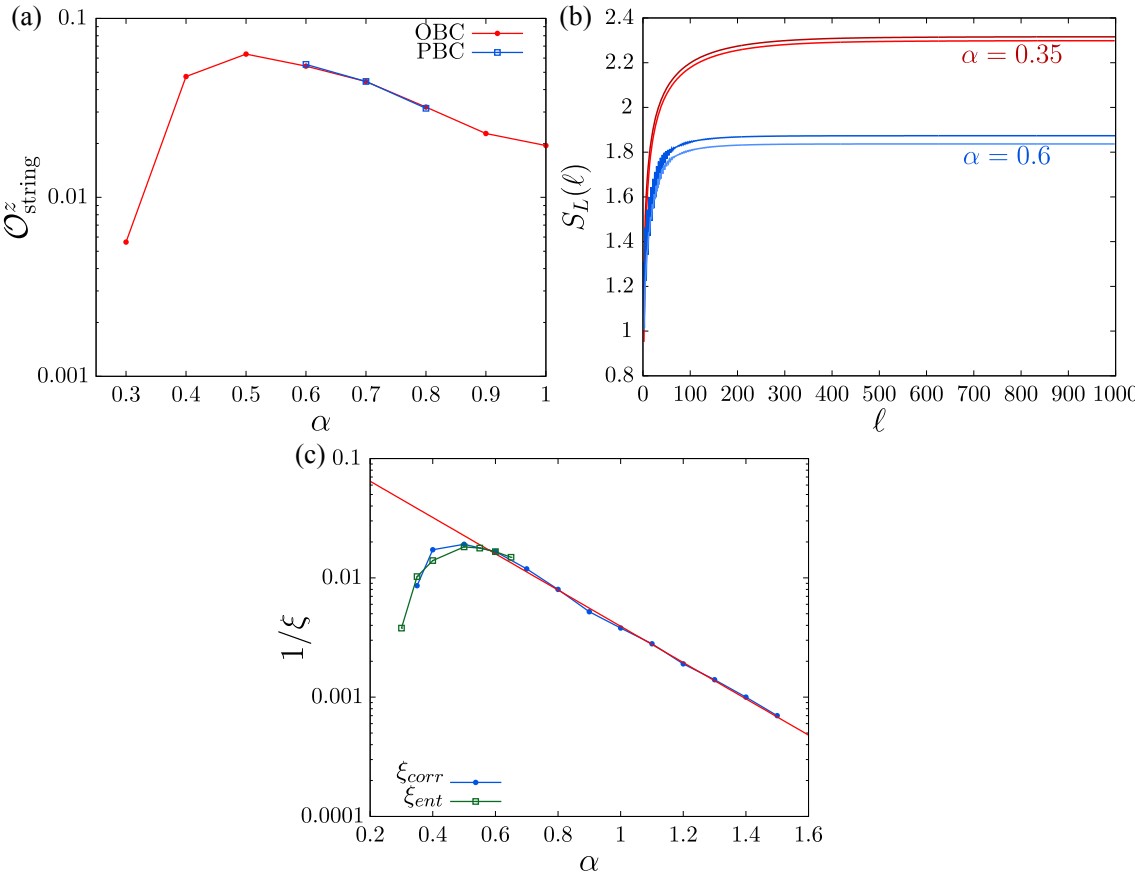

Figure 4: (a)String-order parameter as a function of $\alpha$. Red (blue) line refers to open (periodic) boundary conditions. (b) Entanglement entropy as a function of the subsystem length $l$. (c) Inverse of the decay length estimated from the spin-spin correlation ($\xi_{\text{corr}}$) and the entanglement entropy ($\xi_{\text{ent}}$) as a function of $\alpha$. Red line is a fit with the exponential function $1/\xi_{\text{corr}} = 0.13 \exp(-0.35\alpha)$.

(see App. A). The two-fold degeneracy due to the FM dimerization of the ground state is lifted under OBC and the value of $\mathcal{O}^z_{\text{string}}$ is different for even and odd $j$ ($k$). We thus take their average obtained with $(k, j) = \left(\frac{L}{4}, \frac{3L}{4}\right)$ and $(k, j) = \left(\frac{L}{4} + 1, \frac{3L}{4} - 1\right)$. We confirm the validity of this method by checking the agreement of the OBC results with those obtained under periodic boundary conditions keeping $|k - j| = \frac{L}{2}$. In Fig. 4(a) the string order parameter in the thermodynamic limit is plotted as a function of $\alpha$. The finite value of $\mathcal{O}^z_{\text{string}}$ suggests the formation of a VBS state with a hidden topological long-range order. The string order vanishes when approaching $\alpha = \frac{1}{4}$, indicating a second-order phase transition at the FM critical point. With increasing $\alpha$, it goes through a maximum at $\alpha \simeq 0.55$, which is roughly consistent with the maximum position of the spin gap, and tends slowly towards zero in the limit $\alpha \to \infty$. The maximum value $\mathcal{O}^z_{\text{string}} \sim 0.06$ is much smaller than $\mathcal{O}^z_{\text{string}} = \frac{4}{9} \simeq 0.4444$ for the *perfect* VBS state for the AKLT model [27] and $\mathcal{O}^z_{\text{string}} \simeq 0.3743$ for the $S = 1$ Heisenberg chain [29]. This means that our VBS state is very fragile which is a reason why it is so difficult to detect the spin gap numerically.

Furthermore, the criticality of a 1D system can be definitely identified by its entanglement structure. We use the von Neumann entanglement entropy of the subsystem with length $l$, $S_L(l) = -\text{Tr}_l \rho_l \log \rho_l$, where $\rho_l = \text{Tr}_{L-l}\rho$ is the reduced density matrix of the subsystem and $\rho$ is the full density matrix of the whole system. A gapped state is characterized by a saturation of $S_L(l)$ as as function of $l$ [30]. In Fig. 4(b) the entanglement entropy is plotted as a function

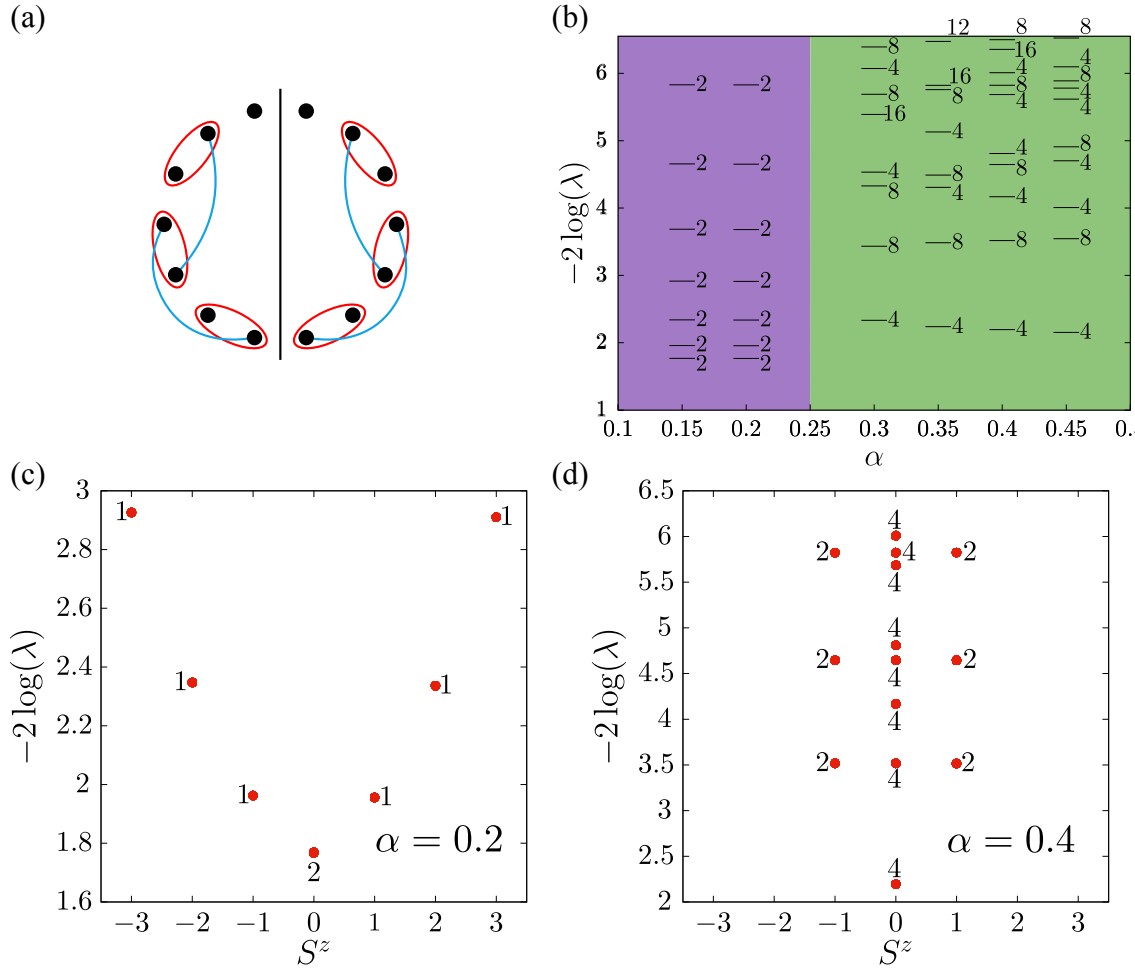

Figure 5: (a) Depiction of the cutting of the system with PBC. Red ellipses represent effective $S = 1$, blue lines represent singlet formation between third-neighbors. (b) ES as a function of $\alpha$, lilac area shows the FM region, green one is the $\mathcal{D}_3$-VBS state. $\lambda$ are the eigenstates of $\rho_\ell$. (c)(d) ES as a function of $S^z$ for (c) $\alpha = 0.2$ (FM) and (d) $\alpha = 0.4$ ($\mathcal{D}_3$-VBS).

of $l$ with fixed whole system length $L = 2000$. We can clearly see the saturation behavior indicating a gapped ground state. The saturation value is slightly split depending on whether the system is divided inside or outside the effective $S = 1$ site. In a VBS state, $S_L(l)$ approaches the saturation value $S_L^{\text{sat}}$ exponentially, i.e., $S_L(l) \sim S_L^{\text{sat}} - a \exp(-l/\xi_{\text{ent}})$; while, the spin-spin correlation decays with distance exponentially, i.e., $|\langle S_0^z S_r^z \rangle| \sim b \exp(-r/\xi_{\text{corr}})$ [31]. For the AKLT VBS state $\xi_{\text{ent}}$ and $\xi_{\text{corr}}$ must coincide, which is indeed what we observe numerically in the $\mathcal{D}_3$-VBS state. [see Fig. 4(c)] For technical reasons, we could determine the spin gap only for $\alpha \leq 0.85$. However, since $\xi_{\text{corr}} \cdot (\Delta/J_2) = \text{const.}$ is expected in the large $\alpha$ regime, a tiny but finite gap is expected up to $\alpha = \infty$.

To further support the existence of topological order in our system, we computed the ES for several value of $\alpha$ through the FM critical point. We studied systems of size $L = 82$ with applying periodic boundary conditions (PBC). We assumed that the system consists of $L = 4n + 2$ sites and it is divided in half as in Fig. 5(a). Since each subsystem includes an odd number of sites, the edge spin state can be directly observed. The results are plotted as a function of $\alpha$ in Fig. 5(b). The FM state ($\alpha < \frac{1}{4}$) has only double degenerate states. The double degenerate state indicates a trivial state because of the area law acting on a periodic system cut at two

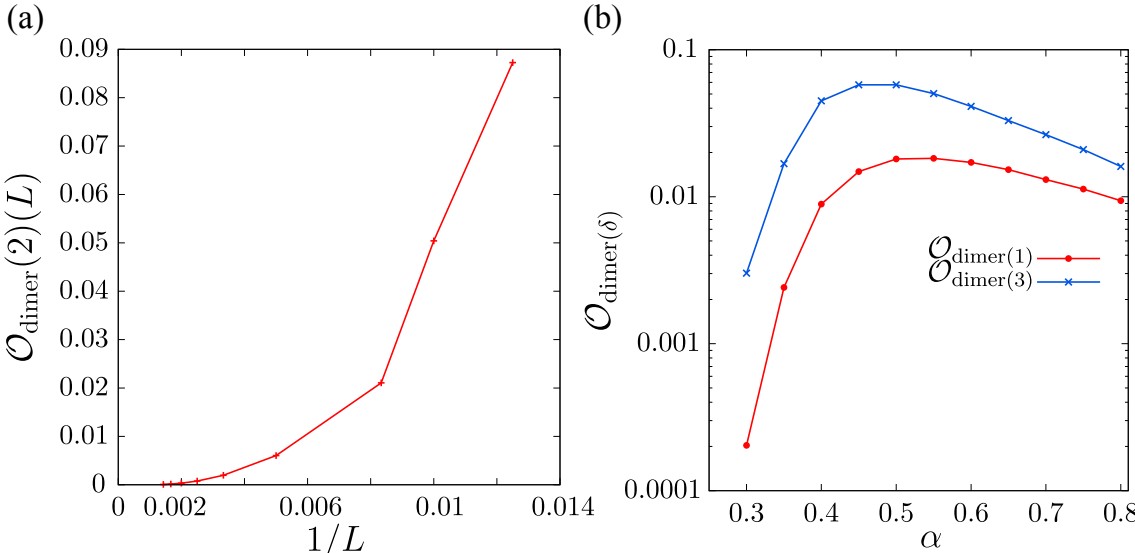

Figure 6: (a) Finite-size scaling of the dimer order parameter for NNN bonds ($\delta = 2$) at $\alpha = 0.6$. The order parameter is vanishing in the thermodynamical limit. (b) Dimer order parameters for NN ($\delta = 1$, red line) and third-neighbor ($\delta = 3$, blue line) bonds as a function of $\alpha$.

points (the typical 1- 3- degeneracy is not possible due to the impossibility of forming a triplet state, having an odd number of spins). The Haldane phase is thus characterized by a four-fold degeneracy of the entire ES [32]. In fact, our $\mathcal{D}_3$-VBS shows $4n$-degeneracy in the entire ES. Therefore, we confirmed that our $\mathcal{D}_3$-VBS state is an expression of the symmetry protected Haldane state. In Fig. 5(c)(d), we show the ES as a function of total spin $z$-component of the subsystem $S^z$ for the FM ($\alpha = 0.2$) and $\mathcal{D}_3$-VBS ($\alpha = 0.4$) states: While in the FM state the double degeneracy is lifted for $S^z \neq 0$ and the spectrum moves away from 0 symmetrically with increasing the Schmidt value, in the $\mathcal{D}_3$-VBS state the Schmidt values are $2n$-degenerate and the spectrum is dense around $S^z = 0$ due to the possibility that the free spins in the two subsystems are aligned ($S^z = 0$) or anti-aligned ($S^z = 1$) .

## 5   Dimerization order

The above analysis makes clear that a gap opens due to the formation of a topologically ordered VBS state but it is not yet obvious how the VBS structure is formed. We can determine a more specific VBS structure by considering the possibility of longer-range dimerization orders. The dimerization order parameter between sites distant $\delta$ is defined as

$$\mathcal{O}_{\text{dimer}}(\delta) = \lim_{L \to \infty} |\langle \mathbf{S}_{i-\delta} \cdot \mathbf{S}_i \rangle - \langle \mathbf{S}_i \cdot \mathbf{S}_{i+\delta} \rangle|, \tag{5}$$

where we take $i = L/2$ for $\delta = 1$ and $i = L/2 - 1$ for $\delta = 2, 3$ (the extrapolated value in the thermodynamic limit does not depend on these choices). If $\mathcal{O}_{\text{dimer}}(\delta)$ is finite for $\delta$, it signifies a long-range dimerization order associated with translational symmetry breaking to period of $4 - 2(\delta \bmod 2)$ [1]. For the case of $\delta = 2$, $\mathcal{O}_{\text{dimer}}(2)$ goes to zero in the thermodynamic limit,

---

[1]This formula becomes obvious for the $\delta = 1$ case: A dimerized bond and an undimerized bond appear alternately along the $J_1$ chain, meaning the symmetry breaking period is 2. For odd values of $\delta > 1$, considering the ladder representation as in Fig. 1(d), the mirror symmetry between the two $J_2$ chains is broken and the translational symmetry along the $J_2$ chains is preserved. It leads to symmetry breaking with period 2 along the $J_1$ chain.

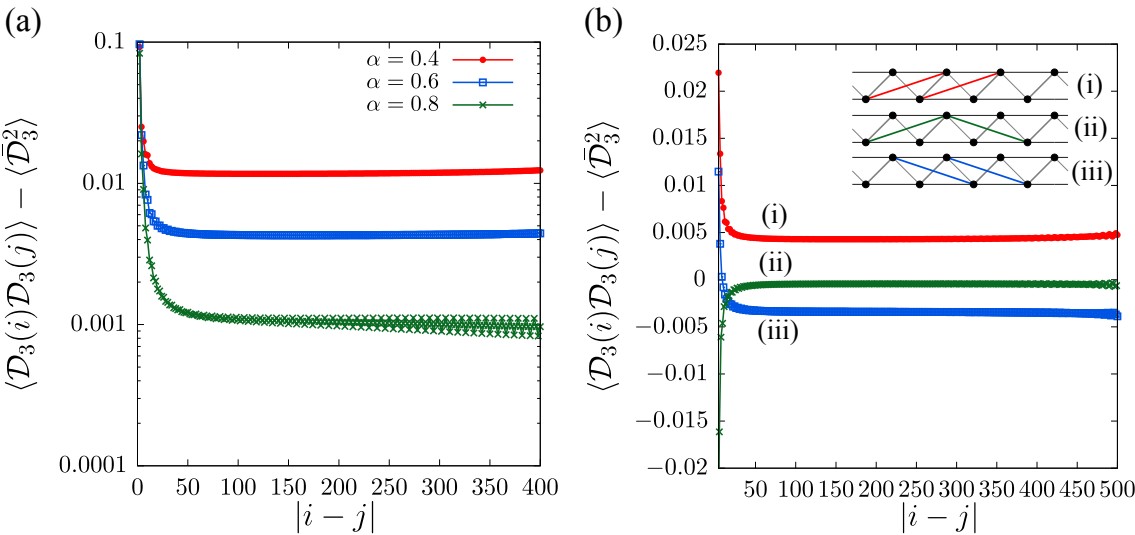

Figure 7: (a) Dimer-dimer correlation $\langle \mathcal{D}_3(i)\mathcal{D}_3(i)\rangle$ as a function of distance $|i-j|$ for several values of $\alpha$. To see the net correlation the product of expectation values $\langle \bar{\mathcal{D}}_3 \rangle^2$ is subtracted. (b) Dimer-dimer correlation functions for the three different kinds of third-neighbor bonds pairs at $\alpha = 0.6$.

as seen in Fig. 6(a). This clearly indicates the absence of long-range dimerization order along the two $J_2$ chains like in Fig. 1(c). Thus, this VBS state can be excluded as a candidate for the ground state for the FM $J_1$-$J_2$ chain. Hence, the possibility of a VBS state with dimerization along two $J_2$ chains is excluded. Whereas for $\delta = 1$ and 3, $\mathcal{O}_{\mathrm{dimer}}(\delta)$ is finite. In Fig. 6(b) the values of $\mathcal{O}_{\mathrm{dimer}}(1)$ and $\mathcal{O}_{\mathrm{dimer}}(3)$ in the thermodynamic limit are plotted as a function of $\alpha$. Remarkably, $\mathcal{O}_{\mathrm{dimer}}(3)$ is significantly larger than $\mathcal{O}_{\mathrm{dimer}}(1)$ despite the longer distance. Moreover, though FM dimerization between fifth-neighbors and AFM dimerization betweem seventh-neighbor may be finite, we expect them to be much smaller than the values reported in Fig. 6(b). We also find that $\langle \mathbf{S}_i \cdot \mathbf{S}_{i+3} \rangle$ is always negative at $\alpha > \frac{1}{4}$ suggesting that a VBS ground state with third-neighbor valence bonds, i.e., $\mathcal{D}_3$-VBS state, is stabilized as shown in Fig. 1(d).

In order to further prove the $\mathcal{D}_3$-VBS picture, we calculate the dimer-dimer correlation function defined as

$$\langle \mathcal{D}_3(i)\mathcal{D}_3(j)\rangle - \langle \bar{\mathcal{D}}_3 \rangle^2, \tag{6}$$

where $\mathcal{D}_3(i) = \mathbf{S}_i \cdot \mathbf{S}_{i+3}$ is spin-spin correlation between the third-neighbor sites $(i, i+3)$ and $\langle \bar{\mathcal{D}}_3 \rangle$ is the averaged value of $\mathcal{D}_3(i)$ over $i = 1, \cdots, L$ in the thermodynamic limit. In Fig. 7(a) we show the dimer-dimer correlation is plotted as a function of the distance $|i-j|$ for different values of $\alpha$. For all $\alpha$ values a fast saturation with the distance is clearly seen. This directly evidences the presence of the long-range $\mathcal{D}_3$-VBS order. In fact, in Fig. 7(a) only the correlations for dimer pairs forming valence bond as in Fig. 7(b)(i) are shown. It would be informative to see the correlation between the other third-neighbor bond pairs. As expected, the correlation between third-neighbor pairs without valence bond saturates to a negative value [Fig. 7(b)(iii)] and that between third-neighbor pairs with and without valence bond vanishes [Fig. 7(b)(ii)].

Thus, the finite spin gap is related to the emergent spin-singlet formation on every third-neighbor bond. To test this concept, we introduce an explicit AFM exchange interaction

For even values of $\delta$, as depicted in Fig. 1(c), the translational symmetry is broken on the $J_2$ chain with a twofold structure, and the mirror symmetry between the two $J_2$ chains is also broken. This leads to a symmetry breaking period of 4 along the $J_1$ chain.

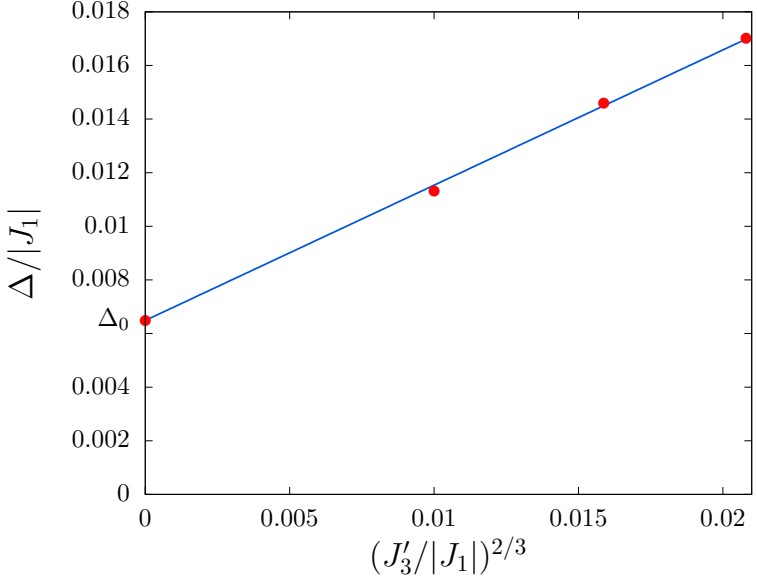

Figure 8: Spin gap $\Delta$ as a function of the third neighbor AFM interaction $J_3'$ for $\alpha = 0.6$. The red line points are data points, the blue line is a linear fitting. We indicate $\Delta(J_3' = 0)$ as $\Delta_0$. The fitting function yields $\Delta - \Delta_0 \simeq 0.5046 J_3'^{2/3}$.

$J_3' \mathbf{S}_i \cdot \mathbf{S}_{i+3}$ on the third-neighbor bonds [see Fig. 1(a)]. Note that $i$ is chosen to be either even or odd depending on the symmetry breaking pattern; in our open chain $i$ is taken to be even. The dependence of $\Delta$ on $J_3'$ with fixing $\alpha = 0.6$ is shown in Fig. 8(a). We find that the spin gap is smoothly enhanced by the AFM $J_3'$. This means that our ground state is adiabatically connected to an explicit formation of the third-neighbor VBS state by $J_3'$. With increasing $J_3'$ the gap increases like $\Delta - \Delta(J_3' = 0) \propto J_3'^{\frac{2}{3}}$, though small but finite intrinsic dimerization should exist at $J_3' = 0$. This is qualitatively the same behavior as in the spin-Peierls transition of the $S = \frac{1}{2}$ dimerized Heisenberg chain [33]. We thus conclude that the ground state of the system (1) is the $\mathcal{D}_3$-VBS state depicted in Fig.1(c). If we regard the system (1) as a diagonal ladder with effective $S = 1$ rungs as in Fig.1(c), the $\mathcal{D}_3$-VBS state may be interpreted as a symmetry protected state [1] with a *plaquette* unit including two effective $S = 1$ rungs, i.e., four $S = \frac{1}{2}$ sites. The plaquette is sketched in the inset of Fig. 9(a). The third-neighbor valence bond is locally stabilized in a $|\sum_{i=1}^{4} \mathbf{S}_i| = 1$, i.e., $S^{\text{tot}} = 1$, sector. The spin gap can be qualitatively estimated from the excitation energy to a state with $|\sum_{i=1}^{4} \mathbf{S}_i| = 2$, i.e. $S^{\text{tot}} = 2$, sector which is projected out from the ground state as in the AKLT model. We plot the excitation energy as a function of $\alpha$. We can see that the tendency of $\Delta$ is qualitatively reproduced by the single plaquette: With increasing $\alpha$, the gap starts to increase at $\alpha = \frac{1}{4}$, goes through the maximum at $\alpha = 0.5$, and then decreases slowly at larger $\alpha$. Moreover, in the $S^{\text{tot}} = 1$ sector the antiferromagnetic spin-spin correlation between sites 1 and 4 is much stronger than that between sites 1 and 3 for $\alpha > 1/4$. This clearly indicates a spin-singlet formation between sites 1 and 4, which corresponds to the third-neighbor valence bond in our $\mathcal{D}_3$-VBS state. Each of the remaining two $S = 1/2$ spins on sites 2 and 3 forms another spin-singlet with a $S = 1/2$ spin in the neighboring plaquette.

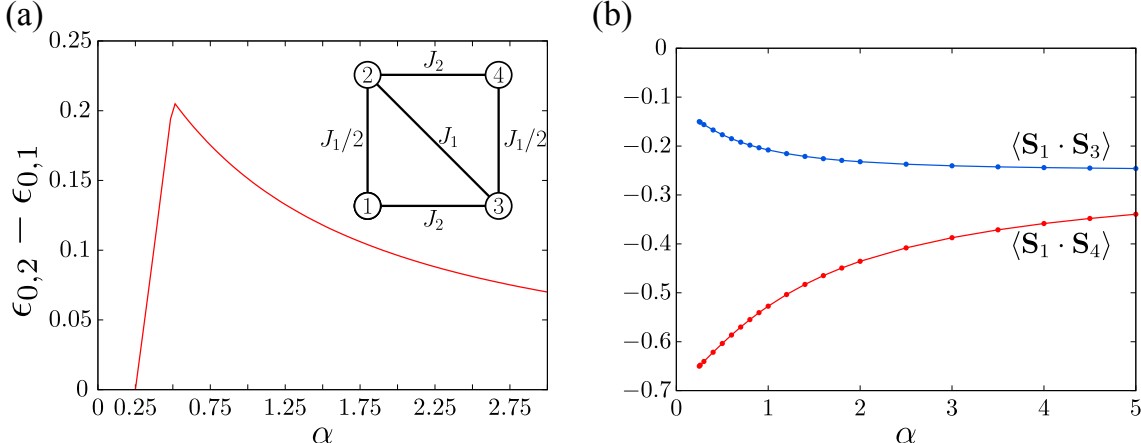

Figure 9: (a) Excitation energy from the $S^{\text{tot}} = 1$ ($\epsilon_{0,1}$) to $S^{\text{tot}} = 2$ ($\epsilon_{0,2}$) sectors in a single plaquette extracted from the diagonal ladder [Fig. 1(c)]. A spin-singlet is formed between sites 1 and 4 in the $S^{\text{tot}} = 1$ sector. (b) Spin-spin correlations in a single plaquette as a function of $\alpha$.

## 6 Matrix product state

Our VBS wave function can be expressed as the matrix product state

$$|\text{VBS}\rangle = \frac{1}{\sqrt{2}}\left[ \text{Tr}\prod_{i \text{ odd}} g_i + \text{Tr}\prod_{i \text{ even}} g_i \right] \tag{7}$$

with

$$g_i = \begin{pmatrix} 0 & 1 \\ -1 & 0 \end{pmatrix} \begin{pmatrix} |\uparrow\rangle_{i+1}|\uparrow\rangle_i & |\uparrow\rangle_{i+1}|\downarrow\rangle_i \\ |\downarrow\rangle_{i+1}|\uparrow\rangle_i & |\downarrow\rangle_{i+1}|\downarrow\rangle_i \end{pmatrix}, \tag{8}$$

where $|a\rangle_{i+1}|b\rangle_i$ ($a, b = \uparrow, \downarrow$) denotes the spin state of effective $S = 1$ site created by the original two $S = \frac{1}{2}$ sites ($i, i+1$). This is similar to the ground state of the AKLT model but the symmetric operation between two spin-$\frac{1}{2}$'s within the effective $S = 1$ site, i.e., $\frac{1}{\sqrt{2}}(|\uparrow\rangle|\downarrow\rangle + |\downarrow\rangle|\uparrow\rangle)$, is not explicitly included (see also App. B). Alternatively, two terms in Eq. (12) correspond to two-fold degenerate states. The Lieb-Schultz-Mattis theorem is thus satisfied. A schematic picture of either one is shown in Fig.1(d), in which every site forms a singlet pair with the third neighbor site. In fact, setting $J_1^{\text{edge}} = 0$ corresponds to an *explicit* replacement of $S = 1$ spin at the each end by $S = \frac{1}{2}$ spin in our effective $S = 1$ chain [34]. It removes the degeneracy due to the edge spin state and enables us to calculate the spin gap with the DMRG method. The essential physics of our $\mathcal{D}_3$-VBS state can be explained by extracting a single *plaquette* including two effective $S = 1$ sites, i.e., four $S = \frac{1}{2}$ sites, in the same way that a combined spin-2 state is projected out in the AKLT model.

## 7 Conclusion

We studied the frustrated FM $J_1$-$J_2$ chain using the DMRG technique. Based on the results of string order parameter, dimerization order parameters, dimer-dimer correlation function, and entanglement entropy, we find a second order phase transition at $\alpha = \frac{1}{4}$ from a FM state to a third-neighbor VBS state with the AKLT-like topological hidden order. This provides a simple realization of coexistence of spontaneous symmetry breaking and topological order, or rather, topological order caused by spontaneous symmetry breaking. It may be helpful to consider this

transition in two steps: (i) *The system exhibits a spontaneous nearest-neighbor FM dimerization, i.e., breaking of translational symmetry, as a consequence of the quantum fluctuations typical of magnetic frustration – order by disorder.* (ii) *By regarding the ferromagnetically dimerized spin-$\frac{1}{2}$ pair as a spin-1 site, the system is effectively mapped onto a $S = 1$ Heisenberg chain and topological order as in the Haldane state is possible. The coexistence of symmetry breaking and topological order is thus allowed. Then, we proposed the third-neighbor valence bond formation as the origin of the finite spin gap since the FM dimerization alone does not lead to a finite gap. The third-neighbor valence bond formation is consistent with the Haldane state with valence bond formation between nearest-neighbor $S = 1$ sites, as the two third-neighbor spins in the $J_1$-$J_2$ chain can be seen as nearest-neighbor spin-1 sites on the effective $S = 1$ chain.* The emergence of third-neighbor VBS formation was also confirmed by the observation of adiabatic connection of the ground state to an enforced third-neighbor dimerized state. Originated from the VBS state, the spin gap opens at $\alpha = \frac{1}{4}$ and reaches its maximum $\Delta \simeq 0.007|J_1|$, which is about two orders of magnitude smaller than that for the AFM $J_1$-$J_2$ chain, at $\alpha \simeq 0.6$. Since the correlation length of spin-spin correlation seems to diverge at $\alpha = \infty$, a tiny but finite spin gap may be present up to $\alpha = \infty$. A typical value for $J_1$ in cuprates is $J_1 = -200$K, which leads to a gap closing at external magnetic field $\simeq 1$ T. In real materials, the exchange couplings have been estimated to be $J_1 = -6.95$meV, $J_2 = 5.20$meV ($\alpha = 0.75$) for LiCuVO$_4$ [35]; $J_1 = -6.84$meV, $J_2 = 2.46$meV ($\alpha = 0.36$) for PbCuSO$_4$(OH)$_2$ [36]. If experimental measurements are performed at very low temperature, a spin excitation gap with magnitude $\Delta = 0.035$meV and $\Delta = 0.013$meV could be observed, respectively.

## Acknowledgements

We thank U. Nitzsche for technical assistance. C.E.A. thanks R. Ray for fruitful discussions.

**Funding information** J. v. d. B. and S. N. are supported by SFB 1143 of the Deutsche Forschungsgemeinschaft.

## A    Derivation of the string order parameter for numerical calculations

The string order parameter for a spin $S = 1$ chain is defined as

$$\mathcal{O}^z_{\text{string}} = -\lim_{|k-j|\to\infty} \langle (\tilde{S}^z_k) \exp(i\pi \sum_{l=k+1}^{j-1} \tilde{S}^z_l)(\tilde{S}^z_j) \rangle, \qquad (9)$$

where $\tilde{S}^z_i$ is the $z$-component of a spin-1 operator at site $i$. In our system, the resultant spin of two $S = 1/2$ spins forming a spin-triplet pair is regarded as an effective $S = 1$ spin. Hence, Eq. (9) can be rewritten in term of $S = 1/2$ spins as

$$\mathcal{O}^z_{\text{string}} = -\lim_{|k-j|\to\infty} \langle (S^z_k + S^z_{k+1}) \exp(i\pi \sum_{l=k+2}^{j-1} S^z_l)(S^z_j + S^z_{j+1}) \rangle, \qquad (10)$$

where $S^z_i$ is the $z$-component of a spin-1/2 operator at site $i$. Considering that the $z$-component of a spin-1/2 spin can only take the values $S^z = \pm 1/2$, we have

$$\exp(i\pi S^z_l) = i\sin(\pm\pi/2) = \pm i,$$

since $\cos(\pm\pi/2) = 0$. Taking pairs of spins $S_l^z S_{l+1}^z$ (within an effective spin-1 site), we get a relation

$$\exp[i\pi(S_l^z + S_{l+1}^z)] = -4S_l^z S_{l+1}^z,$$

where the coefficient 4 accounts for renormalizing the 1/4 factor from multiplying two spin-1/2's. Finally, we obtain a simplified string order parameter:

$$\mathcal{O}_{\text{string}}^z = -\lim_{|k-j|\to\infty}(-4)^{\frac{j-k-2}{2}}\langle(S_k^z + S_{k+1}^z)\prod_{l=k+2}^{j-1}S_l^z(S_j^z + S_{j+1}^z)\rangle, \tag{11}$$

which is expressed only by products of $S^z$.

## B   Matrix product expression of the $\mathcal{D}_3$-VBS state

The $\mathcal{D}_3$-VBS wave function is expressed as a matrix product state

$$|\text{VBS}\rangle = \frac{1}{\sqrt{2}}\left[\text{Tr}\prod_{i\text{ odd}}g_i + \text{Tr}\prod_{i\text{ even}}g_i\right] \tag{12}$$

with

$$g_i = \begin{pmatrix} 0 & 1 \\ -1 & 0 \end{pmatrix}\begin{pmatrix} |\uparrow\rangle_{i+1}|\uparrow\rangle_i & |\uparrow\rangle_{i+1}|\downarrow\rangle_i \\ |\downarrow\rangle_{i+1}|\uparrow\rangle_i & |\downarrow\rangle_{i+1}|\downarrow\rangle_i \end{pmatrix}, \tag{13}$$

where $|a\rangle_i|b\rangle_i$ ($a, b = \uparrow, \downarrow$) denotes the spin state of the effective $S = 1$ site created by the original $S = \frac{1}{2}$ sites ($i, i+1$). Let us perform a part of the product between two effective $S = 1$ sites:

$$\begin{pmatrix} |\uparrow\rangle_{i+1}|\uparrow\rangle_i & |\uparrow\rangle_{i+1}|\downarrow\rangle_i \\ |\downarrow\rangle_{i+1}|\uparrow\rangle_i & |\downarrow\rangle_{i+1}|\downarrow\rangle_i \end{pmatrix}\begin{pmatrix} 0 & 1 \\ -1 & 0 \end{pmatrix}\begin{pmatrix} |\uparrow\rangle_{i+3}|\uparrow\rangle_{i+2} & |\uparrow\rangle_{i+3}|\downarrow\rangle_{i+2} \\ |\downarrow\rangle_{i+3}|\uparrow\rangle_{i+2} & |\downarrow\rangle_{i+3}|\downarrow\rangle_{i+2} \end{pmatrix}$$
$$= \begin{pmatrix} |\uparrow\rangle_{i+1}|\uparrow\rangle_{i+2} & |\uparrow\rangle_{i+1}|\downarrow\rangle_{i+2} \\ |\downarrow\rangle_{i+1}|\uparrow\rangle_{i+2} & |\downarrow\rangle_{i+1}|\downarrow\rangle_{i+2} \end{pmatrix}\otimes(|\uparrow\rangle_i|\downarrow\rangle_{i+3} - |\downarrow\rangle_i|\uparrow\rangle_{i+3}). \tag{14}$$

A spin-singlet is formed between $S = 1/2$ spins at sites $i$ and $i + 3$, namely, between third-neighbor sites. Since the resultant $2 \times 2$ matrix has the same form as before, this matrix product state can be extended up to an arbitrary length.

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
