# Peer review of "Coexistence of valence-bond formation and topological order in the Frustrated Ferromagnetic $J_1$-$J_2$ Chain"

_SciPost Physics, doi:SciPost Phys. 6, 019 (2019)_

## Round 3 · Referee Report · Anonymous (Referee 1) · 2018-10-12

Strengths

1- solid numerical investigation of the ferromagnetic Heisenberg chain with S=1/2 supplemented by frustrating antiferromagnetic coupling between next-nearest neighbors. 2-leading edge DMRG code 3-nice results on the tiny spin gaps in this system. 4- kind of VBS ordering is sorted out.

Weaknesses

1- layout: unclear distinction between main text, appendices and supplements 2- very short, cryptic explanations 3- slight overselling of the results 4- explanation of power law with exponent 2/3 unclear 5- some spelling errors in the references

Report

The authors present a solid numerical investigation of the ferromagnetic Heisenberg chain with S=1/2 supplemented by frustrating antiferromagnetic coupling between next-nearest neighbors. They use a leading edge DMRG code to obtain nice results on the tiny spin gaps in this system. In addition, they sort out which kind of ordering occurs. It appears to be a third-neighbor dimerization. All in all, the system behaves qualitatively similar to a Haldane chain and a AKLT chain.

The topic of unexpected ordering is timely and the results are nice so that publication in indicated. However the presentation and the layout of the manuscript needs some improvement:

(i) There are a number of appendices. Are they appendices or will they appear as supplement? The references to them in the main text should be consistently to appendices or to supplements.

(ii) Often, equations and figures are quoted in the main text which appear only in the appendices. This is a nuisance for the readers. All material actively used in the main text should be presented in the main text - there are no limits on length in SciPost anyway. - Why is Eq. 3 on two lines? - Which Eq. 8 is simplified? - Why is the definition of the string order not a proper equation with number?

(iii) Sometimes the presentation is too short and too cryptic. Again, no length limit constrains the authors to improve on comprehensibility, for instance: - What is meant by "arbitrary set of valence bonds" (end of page 3)? - Which "two-fold degeneracy" is referred to after Eq. (3)? - Where does the formula 4-2(delta mod 2) come from? (line 1 on page 5)

(iv) Understandably, the authors want to sell their results. But I think they exaggerate slightly: - Why is the phenomenon "order by disorder"? That term usually refers to classical degeneracy lifted by fluctuations. I do not see this here. And if it were, it were not the "first example" as stated in the abstract, see Haldane chain or Majumdar-Ghosh chain. - Clearly, the system shows long range string order. But why is this topological? What is the relevant manifold, the fibre bundle and the topological invariant of this topological order? If the authors insist on the term "topological" these issues should be discussed.

(v) In the Introduction, a number of compounds is given which potentially realize the investigated chain. The reader will wonder what one can learn from the theoretical results for these compounds. So it should be discussed briefly which messages for the compounds can be taken from the theoretical results.

(vi) The dependence of the spin gap on J_3' with a power law with exponent 2/3 is very interesting. But the references quoted treat the massless case without gap in absence of VBS. Hence their results do not apply here. I presume the exponent can rather be explained by the binding of two solitons. The authors may consider this scenario, see for instance Uhrig et al. EPJB 7, 67 (1999).

(vii) There are spelling errors in the references, see e.g. Refs. 6, 21 and 24.

Requested changes

1-Please address A-G consistently as appendices or as supplement.
In the latter case, they should be submitted as separate file.
2-Eqs and Figs. quoted in the main text should be given in the main text.
3-Too short and cryptic statements should be expanded, see examples in Report
4-Avoid the expression "order by disorder". It is in appropriate here.
5-The last sentence of the abstract should be omitted.
6-Either avoid the term "topological order" for string order or explain
the topological aspect in more detail.
7-Briefly mention any relevance of the theoretical results for the
compounds quoted in the beginning.
8-Explain the exponent 2/3 for gapped solitons, not gapless solitons.
9-Check list of references for spelling errors.

  • validity: good
  • significance: good
  • originality: good
  • clarity: ok
  • formatting: below threshold
  • grammar: good

Author:  Cliò Efthimia Agrapidis  on 2018-11-22  [id 354]

(in reply to Report 1 on 2018-10-12)
Category:
answer to question

We thank the referee for his/her comments. We here reply to them.

(i) In the new version, we changed the main text to refer to the appendices as, for example, App. A. We now moved
most of the appendixes to the main text, apps. B-G.

(ii) We now moved materials addressed in the main text to the main text itself. We put eq. 3 in one line and the simplification of the string order parameter is presented as a proper equation with number. The reference to eq. (8) in the main text has been corrected.

(iii)
-The statement "arbitrary set of valence bonds" refers to the possibility that the VB might form in different directions, as shown in fig. 1(b). We added further explanations about this in the main text.
-We refer to the two-fold degeneracy due to the dimerization: because of the particular boundary conditions we use (explained in the methods section), the degeneracy is lifted with OBC.
-When $\delta$ is odd the period is 2. It is clear for $\delta=1$ because a dimerized and an undimerized bond appear alternately along the $J_1$ chain. For $\delta>1$ it may be helpful to consider the system within the topologically equivalent ladder geometry. In this case, as seen in Fig.1(c) the translation symmetry along the $J_2$ chains is not broken but the mirror symmetry between two $J_2$ chains is broken. This leads to a period of 2 along the $J_1$ chain. Whereas for even $\delta$, as seen in Fig.9(b), the translation symmetry is broken along the $J_2$ chains with twofold structure and also the mirror symmetry between two $J_2$ chains is broken. Thus, the period is $2\times2=4$ along the $J_1$ chain. We added a more detailed explanation about this in the main text.

(iv)
-Our system is a geometrically frustrated chain: its classical ground state is highly degenerate, the quantum fluctuations lift this degeneracy with the formation of FM dimers and valence bonds. As the referee points out, the Majumdar-Ghosh chain is a consequence of order by disorder but the presence of topological order has not been discussed yet. We expanded the text in the manuscript on this point.
-The nonlocal string order parameter is a well-used quantity to detect the AKLT or Haldane state as a topological state. We appreciate the point raised by the referee and in order to further prove the topological nature of the AKLT-like VBS (Haldane) state appearing in this model, we have computed the entanglement spectra for different values of $\alpha$ to show the existence of fractionalized edge excitations: they clearly show a QPT from a trivial state (FM state) to a non-trivial one ($\mathcal D_3$-VBS state). We attach a figure showing (a) how the system is cut and (b) the resulting ES as a funtion of $\alpha$. We added these results in the main text of the manuscript.

(v) We now discuss the implications of our results on the cited compounds in the conclusions.

(vi) We agree with the suggestion. Since our system is massive, it should be explained by binding of two solitons. We replaced our previous citation about the exponent by Uhrig et al. EPJB 7, 67 (1999). We also added some more explanations about this issue.

(vii) We carefully went through the references and corrected the spelling mistakes.

Attachment:

ES.pdf

---

## Round 3 · Referee Report · Anonymous (Referee 2) · 2018-10-24

Strengths

  1. Accurate DMRG numerics.
  2. Attempt to uncover the tiny spin gap in the FM chain.

Weaknesses

  1. Presentation needs to be improved. Since there is no restriction on space, introduction and conclusion may be expanded and many of the appendices may be integrated with the main text.

  2. The authors need to provide a clear summary pf results and improvements over existing literature to put the work in proper context.

Report

May be published if the authors improve their presentation.

Requested changes

  1. Expand introduction to survey existing results in greater details.

  2. Appendices B-F may be integrated with the main text in appropriate locations.

  3. The conclusion may be expanded and the significance of the numerical results obtained may be discussed in greater details.

  • validity: high
  • significance: good
  • originality: high
  • clarity: low
  • formatting: below threshold
  • grammar: reasonable

Author:  Cliò Efthimia Agrapidis  on 2018-11-22  [id 353]

(in reply to Report 2 on 2018-10-24)
Category:
answer to question

We thank the referee for his/her comments. We here reply to them.

-We integrated the appendices and expanded some parts of the text.

-In the updated manuscript we change part of the presentation of the results, aiming to put them in better context.

---

## Round 3 · Referee Report · Anonymous (Referee 3) · 2018-11-8

Strengths

The problem addressed is a non-trivial one, where for a range of parameters excitation gap above the ground state nearly vanishes. DMRG has been used effectively to show the finite but extremely small small gap. It also brings out existence of string order this nearly gapless system.

Weaknesses

Figuure 1 and 9 shows valence bond order and emergent spin-1 bonds. Except for figure 1d, res of the figures are some what confusing (even though technically correct) and not illuminating. Let the authors come up with better illustrations.

Valence bond order implies existence of soliton excitation namely spinons.
I would have liked the authors to discuss them at least qualitatively. As the gap is nearly vanishing, spinons are nearly deconfined in a wide parameter region. What is the relation of the spinon with edge spin-half excitations ?

Report

It is a good paper that confirms by powerful numerics existence of a tiny gap, string order and valence bond orders.

Requested changes

Better, non cluttered figures will make the discussed orders more transparent.

  • validity: high
  • significance: high
  • originality: good
  • clarity: ok
  • formatting: reasonable
  • grammar: good

Author:  Cliò Efthimia Agrapidis  on 2018-11-22  [id 352]

(in reply to Report 3 on 2018-11-08)
Category:
answer to question

We thank the referee for his/her comments. We here reply to them.

We revised the figures to improve the presentation of the different states.

Thank you for the interesting question to improve the manuscript. As the referee suggests, spinons are nearly deconfined. In Fig.6(a) a spinon is created at the system edges as an edge spin-half excitation. Typically, the Friedel oscillation decays quickly (decay length is of the order of $1$) from the edges in a gapped system. If the edge spin-half is completely free, the decay length is 0. However, in our system it decays very slowly and the amplitude seems to be finite even around the system center for $L=600$. This is also consistent with an exponential decay of the spin-spin correlation with very large decay length, $\xi\sim50$ ($\alpha\sim0.6$) at minimum. We added this explanation in the main text.

---

## Round 4 · Referee Report · Anonymous (Referee 2) · 2018-12-12

Strengths

See earlier report

Weaknesses

See earlier report.

Report

The authors have implemented my suggestions. I feel that the manuscript is in better shape now and can be publsihed.

---

## Round 4 · Referee Report · Anonymous (Referee 1) · 2018-12-14

Strengths

1- solid numerical investigation of the ferromagnetic Heisenberg chain with S=1/2 supplemented by frustrating antiferromagnetic coupling between next-nearest neighbors. 2-leading edge DMRG code 3-nice results on the tiny spin gaps in this system. 4- kind of VBS ordering is sorted out.

Weaknesses

  • slight overselling of the results

Report

The manuscript has improved so that I am inclined to recommend its publication.
However, one of the two main selling points, order by disorder,
are still not well corroborated by the results.
What is the classical degeneracy of the system? I do not see any except for
fine-tuned parameters J1, J2 and J3, but not for generic sets of these parameters.
(The occurrence of a gap in a dimerized AFM chain as such destroys the LRO,
but it does not imply any "order by disorder" mechanism.)

Requested changes

I do not request changes, but I recommend to give the authors
the opportunity to substantiate their one of their main claims, see Report.

---

## Round 4 · Author Response

Dear Editor,
We resubmit our paper titled "Coexistence of valence-bond formation and topological order in the Frustrated Ferromagnetic $J_1$-$J_2$ Chain". We have addressed the referee comments, adding new results on entanglement spectra and performing minor revisions. A list of changes is visible below.

Kind regards,
C. E. Agrapidis, S.-L. Drechsler, J. van den Brink, S. Nishimoto

---

## Round 4 · List of Changes

-Appendices B-C moved to Sec. 3 Spin gap.
-Appendices D-F moved to Sec. 5 Dimerization order.
-Fig. 1 changed to make the represented orders more clear and to include previous Fig. 9(b).
-New Fig. 2 includes the figures from App. B.
-New Fig. 3 is similar to previous Fig. 2: the inset is now Fig. 3(c) and we include the finite size scaling for the spin gap in Fig. 3(b).
-New Fig. 5 shows results for the entanglement spectra of the system.
-New Fig. 6 includes previous Fig. 9(a) and 4(a).
-New Fig. 7 includes previous Fig. 4(b) and 10.
-New Fig. 9 includes previous Fig. 11 and 12.
-More details about the use of the denomination "order by disorder" have been added in the introduction.
-An explanation of the statement "arbitrary set of valence bonds" has been added to the manuscript.
-The cause for the two-fold degeneracy of the ground state, namely the ferromagnetic dimerization, is now explicitly addressed.
-To support our claim of topological order in the $\mathcal D_3$-VBS state, we have included new results about entanglement spectra in Sec. 4 Valence Bond Solid.
-We added a footnote to explain the formula $4-2(\delta \bmod 2)$ for the symmetry breaking period.
-We changed the references to Supplemental Material to be to App. A or App. B
-We added an explanation on how the phase transition works in Sec. 7 Conclusion to better put our findings into context.
-We added a short discussion about the consequences of our theoretical findings for real materials in Sec. 7 Conclusion.

---

## Editorial Decision

published